# ELSA: Evaluating Localization of Social Activities in Urban Streets using Open-Vocabulary Detection

**Maryam Hosseini**[1*]    **Marco Cipriano**[2*]    **Daniel Hodczak**[3]
**Sedigheh Eslami**[2]    **Liu Liu**[1]    **Andres Sevtsuk**[1]    **Gerard de Melo**[2]
[1]Massachusetts Institute of Technology (MIT)
[2] Hasso Plattner Institute (HPI)
[3]University of Illinios Chicago (UIC)
maryamh@mit.edu, marco.cipriano@hpi.de

## Abstract

Existing Open Vocabulary Detection (OVD) models exhibit a number of challenges. They often struggle with semantic consistency across diverse inputs, and are often sensitive to slight variations in input phrasing, leading to inconsistent performance. The calibration of their predictive confidence, especially in complex multi-label scenarios, remains suboptimal, frequently resulting in overconfident predictions that do not accurately reflect their context understanding. The Understanding of those limitations requires multi-label detection benchmarks. Among those, one challenging domain is social activity interaction. Due to the lack of multi-label benchmarks for social interactions, in this work we present ELSA: Evaluating Localization of Social Activities. ELSA draws on theoretical frameworks in urban sociology and design and uses in-the-wild street-level imagery, where the size of social groups and the types of activities can vary significantly. ELSA includes more than 900 manually annotated images with more than 4,000 multi-labeled bounding boxes for individual and group activities. We introduce a novel re-ranking method for predictive confidence and new evaluation techniques for OVD models. We report our results on the widely-used, SOTA model Grounding DINO. Our evaluation protocol considers semantic stability and localization accuracy and sheds more light on the limitations of the existing approaches.

## 1 Introduction

"*For it is interaction, not place, that is the essence of the city and of city life.*"

*(Melvin M. Webber, 1964, 147)*

In recent years, increased focus on the human scale of the cities has drawn more attention to public spaces and pedestrian facilities. For decades, urban scholars from various fields have been fascinated by the complex interplay between public spaces and the social interactions they support [28, 37, 17]. However, traditional scientific inquiry into the distribution of social activities across urban streets have been hampered by high data collection costs and extensive time requirements.

The emergence of advanced computer vision techniques such as object detection and semantic segmentation together with the availability of public sources of street-level imagery have opened

---

*Equal contribution.

Submitted to the 38th Conference on Neural Information Processing Systems (NeurIPS 2024) Track on Datasets and Benchmarks. Do not distribute.

new avenues for conducting comprehensive observational studies at reduced cost and increased scale. Activity recognition techniques are mostly designed to work with videos [23], since, by nature, human activity involves motion and sequence of actions. Yet, acquiring continuous video footage across an entire city over time entails substantial data storage requirements and processing costs, making it very difficult to scale. Object detection on still images emerges as a low-cost, efficient, and applicable method, as it allows for the identification and localization of complex social interactions in diverse settings, where the environmental context significantly influences the range of possible social interactions and where each image can contain a large number of people engaged in diverse activities.

While conventional object detection models are trained in closed-vocabulary settings and rely heavily on predefined classes, open-vocabulary detection (OVD) models aim to transcend traditional object detection models, and utilize the abundance of language data in order to enable the detection of classes with less representation in standard benchmark training data. A robust OVD model is expected to handle a wide range of input terms and phrases that were not explicitly part of its training set. This is crucial for models deployed in real-world settings, such as urban streets, where unpredictable and varied interactions are common. The absence of benchmark data for open-vocabulary detection of social and individual actions in still images 'in the wild' hinders the development of robust models that generalize well across diverse and spontaneous urban scenarios, where the context and variability of human activities are far greater than those typically encountered in controlled environments. Furthermore, OVDs pose new challenges in both localization and semantic understanding of unseen new categories. They often struggle with semantic consistency across diverse inputs, demonstrate sensitivity to slight variations in input phrasing, and the calibration of their predictive confidence, especially in out-of-distribution scenarios, remains suboptimal, resulting in overconfident predictions that do not accurately reflect their actual accuracy [33, 8].

In response to these challenges, we propose ELSA, a new benchmark dataset and evaluation framework in order to evaluate the performance of OVD models in recognizing and localizing human activity in urban streets from still images. We employ a multi-labeling scheme and define 33 unique individual labels regarding human activities. These labels can concurrently be associated to each annotated bounding box. As a result, ELSA includes more than 4,000 bounding boxes annotated with 115 unique combination of human activities for 900 street view images. In order to evaluate the robustness of OVD models, ELSA contains challenging scenes with humans located relatively far from the camera as well as scenes containing pictures of people, which are likely to get falsely detected as genuine people by such models.

Furthermore, due to the close ties of OVD models with language features, using the for evaluation purposes entails certain challenges. We design a novel re-ranking score, namely N-LSE, metric to rank the predicted bounding boxes based on the most salient sub-phrases and tokens of the query, and take into account the token-level correspondence of language with the visual features on the predicated area. We further propose Confidence-Based Dynamic Box Aggregation (CDBA), in order to handle multiple detected predictions of the same object, which overcomes the shortcomings of the Non-Maximum Suppression (NMS) [38] method and its variation NMS-AP.

## 2   Related Work

**Social Interactions in Public Spaces.** Vibrant streets rich in interpersonal exchange have fascinated urban scholars because of their social qualities as well as fundamental indicators of sustainable urban environments [28]. William Whyte [37] along with Jacobs [17] highlight the intrinsic value of public spaces in fostering vibrant social life. Jan Gehl [12] describes activities in the public spaces as a spectrum between optional activities, e.g., talking with friends, and necessary activities, e.g., walking to work. The public space observational method [13] delineates between active social group activities, e.g., dining or talking together, and passive activities, such as strangers sitting on a bench checking their cell-phones. Inspired by this research, we define the target set of social activities in ELSA.

**Open-Vocabulary Object Detection.** OVD, first introduced by Zareian et al. [40], primarily tackles the limitation of traditional object detection models that rely on pre-defined closed set of objects

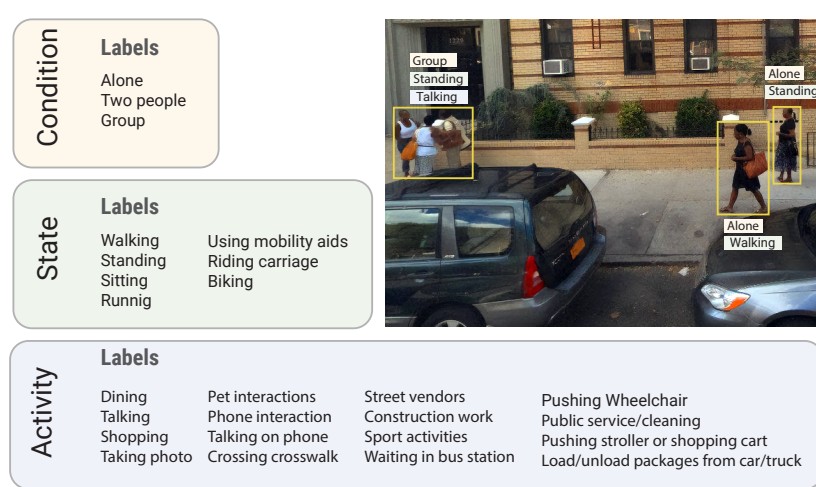

Figure 1: Examples of label space in our social interaction study.

[4, 31, 21] tested on various OVD benchmark datasets [33, 38]. At their core, a vision-language contrastive loss is often used for aligning semantics and concepts in the two modalities [20, 27, 6, 18, 30, 24] with additional soft token prediction in MDETR [20]. Using a dual-encoder-single-decoder architecture, Grounding DINO [27] extends DINO [41] such that given a text prompt, query selection is performed to select the text features more relevant for the cross-modal decoder. A contrastive loss for aligning the output of the decoder and text queries along with a regression L1 loss and generalized union over intersection is optimized end-to-end for the detection.

**OVD Evaluation.** The standard evaluation metric for object detection is the mean of the per-class average precision (mAP) [11]. As shown by Dave et al. [8], standard AP is sensitive to changes in cross-category ranking. Furthermore, [38] shows the inflated AP problem and proposes to suppress that using class-ignored NMS-AP that unifies multiple predictions of the same box and assigns the highest confidence label to that box. Relying on the maximum-logit confidence, this method is also prone to misrepresent the correct ranking of relevant boxes and can inaccurately represent the robustness and stability of the model in predicting the correct class, as it is merely relies on the maximum-logit token from the query.In contrast, our approach ranks the predicted boxes with respect to all tokens in the query, which is crucial for multi-label scenarios.

**Activity Localization Datasets.** Activity localization involves analyzing the activities in a sequence images [2, 3, 10, 42, 42]. A seminal study by Choi et al. [7] focuses on in-the-wild pedestrian action classification from videos. Recent advancements in Zhou et al. [42] and Wang et al. [36]combine appearance and pose data with transformers in order to enhance interaction recognition and improve the detection of complex human behaviors. Li et al. [25] added cognitive depth with the HAKE engine, which uses logical reasoning to analyze human–object interactions. However, all of these models are tested on video datasets such as a volleyball dataset [16], AVA-Interaction [36], HICO-DET [5], V-COCO [15], NTU RGB+D [34, 26], and SBU-Kinect-Interaction [39]. Among previous work, Ehsanpour et al. [10] includes annotated videos of university campus scenes for group-based social activities and enables group-based social activity recognition. In contrast, ELSA aims at localization of social activities in still images, which is a more challenging problem. In image sequences, activities can be recognized based on the object movements across consecutive images. In contrary, for still images, localization models need to infer activities from the snapshot of the moment shown.

# 3 ELSA: Evaluating Localization of Social Activities

## 3.1 Benchmark Dataset

Motivated by the lack of available benchmark data for detection of social interactions and individual activities in still images, we propose ELSA. The goal is to enable the evaluation of state-of-the-art object detection models in detecting various levels and patterns of human activity and interactions. In this section, we provide a detailed description of ELSA and its unique characteristics.

**Image Resources.** We chose New York City as the site of interest, due to the well-known presence of lively streets and public spaces. We compiled street-level images from two different sources: Microsoft Bing Side-view [22] and Google Street View [14, 1]. The Bing imagery provides time-stamps, making it possible to choose days and times with a higher probability of encountering pedestrians on the streets.

**Target Labels** We draw on the literature on active design and urban vibrancy (see Section 2) to select our primary individual labels. ELSA exhibits non-disjoint label spaces, where multiple concurrent labels can be applied to the same object in a multi-labeling scheme that encompasses the nuances of human behavior and context. Labels are grouped into four categories: 1) Condition: defines the social configuration of the subjects as *alone*, *two people*, or *group*. These labels are disjoint and denote mutually exclusive social settings, establishing the primary context for potential interactions, such as solo activities, limited interactions, or group dynamics; 2) State: captures the physical disposition or activity mode of the subjects, such as *walking* or *sitting*. While disjoint for individuals, these labels can co-occur in couple or group scenarios, indicating stationary engagement (*standing*, *sitting*) or transient interactions (*walking*, *biking*); 3) Action: reflects specific behaviors or activities, such as *dining* or *talking*. We report additional information about the label categories in Appendix 6.1.

**Annotation Process.** We customized the open source Label Studio tool [35] for annotation and integrated YOLOv8 [19] for pre-detecting the initial objects. A group of four people manually corrected the initial bounding boxes and annotated the label combinations. Finally, an urban planning expert reviewed the label and bounding box accuracy for all annotations.

Examples of ELSA's annotations are depicted in Figure 1. Additional examples are included in Section 6.2.

**Annotation Cleaning.** After the initial annotations, we performed sanity checks on the disjoint labels and defined a set of sanity rules, e.g., a bounding box with just one person cannot have two contradictory states of sitting and walking at the same time. The full list of these sanity rules are provided in Section 6.3. We applied the sanity rules to all the annotated bounding boxes and re-annotated the ones that did not pass the sanity checks. We repeated this process until all bounding boxes passed our defined sanity rules.

**Dataset Statistics.** ELSA includes 924 images with more than 4.3K annotated bounding boxes for social and individual activities. In total, there exist 34 distinct single labels in ELSA. Since we have a multi-labeling scheme, each bounding box can have 2 or more of the distinct 34 labels associated with it. As a result, ELSA includes 112 unique combinations of human activities. Figure 2 shows the distribution of the distinct labels as well as the distribution of combinations of multiple labels in ELSA.

**Prompt Formation.** Unlike physical objects, activities and human–human or human–object interactions pose significant challenges in being accurately captured by a single word or label. To investigate this, we conducted a series of tests on various models, examining their responses to prompts with verbs like "walking," "talking," or "standing," and phrases like "walking alone" or "talking in groups." As expected, the results were often inaccurate or non-existent. These models require more detailed natural language descriptions to detect these activities correctly, such as "an individual sitting on a bench."

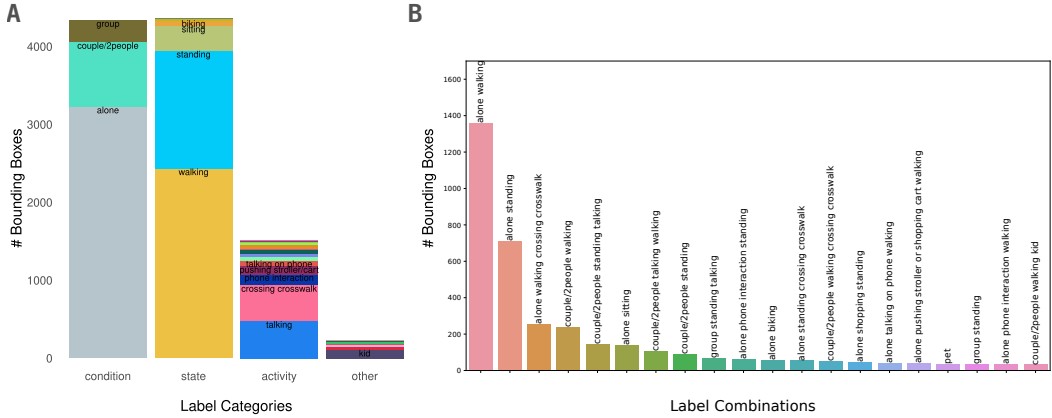

Figure 2: Overview of the distribution of activities in ELSA. (A) Number of bounding boxes per single activity label, (B) Distribution of the top 20 activity labels that occur together.

To address this need, we enhanced ELSA with the ability to generate precise, naturally phrased sentences for each label combination and their near synonyms. This capability ensures that the models receive comprehensive descriptions, significantly improving their detection accuracy.

### 3.1.1 Challenging Scenarios

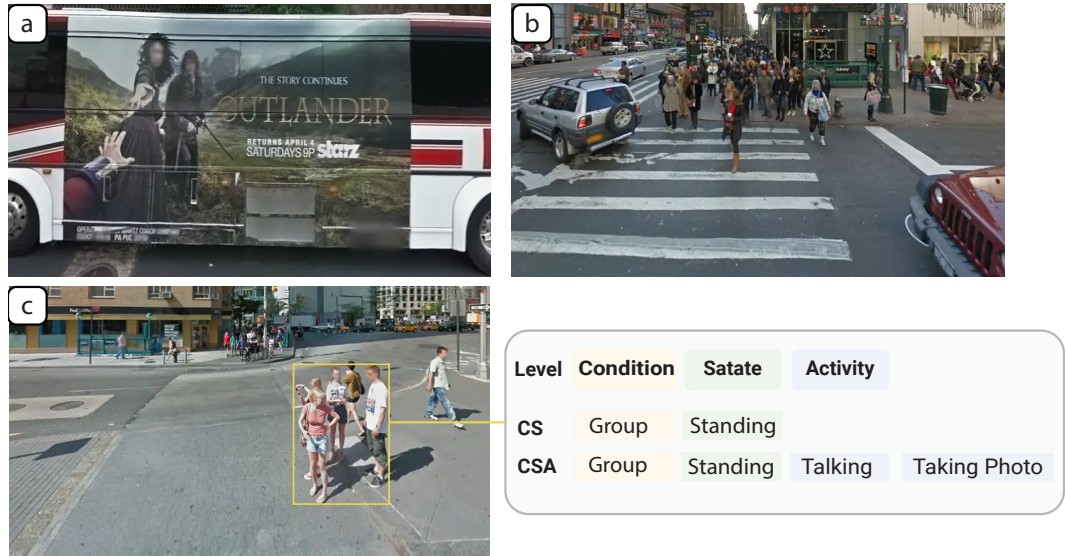

Figure 3: Example of challenging scenarios: a) Printed image of people that are not to be recognized as genuine; b) Crowded scene with people standing at different distances from camera; c) Prompts at two levels *cs* and *csa* for one target in the image.

**Challenging Scenarios.** ELSA includes still images from 'in the wild' scenarios, which examines the robustness and generalizability of the state-of-the-art models across diverse and spontaneous urban scenarios where the context and variability of human activities are far greater than those typically encountered in controlled environments. Thus, ELSA poses two types of challenges for activity recognition and localization models:

1. Challenges in the visual data: ELSA include negative sets in scenes without actual pedestrians but with printed images of people on billboards, buses, or walls (see Figure 3-a), as well as mannequins in store fronts. There are instances of people standing far away from the camera, making them difficult to detect. We also have crowded scenes with obstructions, where detecting all targets can be challenging (see Figure 3-b).

2. Prompt level challenges: We employ a three-level benchmark to increase the complexity of the query prompts at each level. Each level's queries are designed to return all instances of the target label combination that includes these sub-category labels. Label combinations in ELSA follow one of the following patterns: "Condition + State" (CS, e.g., "group standing"), or "Condition + State + Activity" (CSA, e.g., "group standing talking and taking photo") or "Condition + State + Activity + Other" (e.g., "group standing talking and taking photo with coffer or drink in hand") (see Figure 3-c for a two-level prompt example).

## 3.2 Evaluation Approach

One distinguishing factor of open-vocabulary detection is the capability to draw on the natural language features to predict novel classes. This means that in zero-shot prediction, the semantic label of the class (the query phrase) can play a critical role in model performance. Ideally, the model should be able to recognize the details of the main target (semantic understanding), correctly associate near synonyms to the same object with close confidence level (semantic stability), and accurately localize the target in the image by connecting natural language and visual features (localization). All three aspects are important in measuring the performance of OVDs. In this work, we focus on evaluating semantic stability as well as localization capabilities of the OVD models.

### 3.2.1 Re-ranking Predicted Bounding Boxes

In open-vocabulary detection, each predicted bounding box is typically associated with a confidence score and an array of logits. These logits quantify the model's confidence in the relationship between the visual features within the bounding box and specific tokens. Often, the confidence score of a bounding box is determined by the highest logit value, i.e., Max-Logit, among all tokens [27], which usually corresponds to prevalent object classes, such as "person". However, unlike single-object detection, multi-label human activity and interaction detection presents additional challenges for identifying multiple overlapping targets, activities, and interactions within the same scene. Thus, bounding boxes must reflect not only the presence of the targets but also their states and conditions with higher confidence.

In complex multi-label scenarios, the commonly employed Max-Logit approach may not yield the most accurate representations. To address this limitation, we propose a re-ranking approach that effectively considers the logits of all the tokens for deriving the final score. Specifically, we propose considering the Normalized Log-Sum-Exp (N-LSE) function over tokens as:

$$\text{N-LSE}(\mathbf{z}) = \log\left(\frac{1}{T}\sum_{t=1}^{T}e^{z_t}\right) = \log\left(\sum_{i=1}^{T}e^{z_t}\right) - \log(T), \tag{1}$$

Here, $\mathbf{z}$ represents the vector of logits, and $T$ is the number of elements (corresponding to each token) in $\mathbf{z}$. Following previous work [32], our evaluations prune the predicted boxes with an N-LSE of less than 0.3.

### 3.2.2 Confidence-Based Dynamic Box Aggregation (CDBA)

A common issue with OVD models is that they can achieve high Average Precision (AP) by predicting multiple boxes for the same object across different prompts. Yao et al. [38] proposed a variation of non-maximum suppression (C-NMS), which selects the box with the highest confidence as a true positive (TP) and suppresses the rest as false positives (FP). However, this approach has notable drawbacks: 1) It does not reveal the model's vulnerability to making disjoint predictions with similar

confidence levels, and 2) It may incorrectly suppress true positives with confidence levels close to the highest prediction as false positives. To overcome these problems, we propose the *Confidence-based Dynamic Box Aggregation* method (Algorithm 1) to handle overlapping bounding boxes by considering the range of confidence scores within the group, and classifying boxes based on the coherence of predicted prompts. Here, instead of only looking at the maximum prediction confidence, we consider the confidence range of predicted overlapping boxes, and keep the ones close to the maximum (<0.2 difference), while suppressing the rest. Given that our N-LSE-based score threshold is 0.3, the additional 0.2 threshold on the score, at minimum, puts us around the 0.5 margin, which is deemed sufficiently high to be counted as a TP, if matching the ground truth.

---

**Algorithm 1** Confidence-Based Dynamic Box Aggregation (CDBA)

---

1: **Input:** Groups of overlapping bounding boxes $G$ from multiple prompts on a given image
2: **Output:** Classified boxes with adjusted scores
3: **for** each group $g \in G$ **do**
4:     Compute the range of scores $R = \max(\text{Scores}) - \min(\text{Scores})$
5:     **if** $R > 0.20$ **then**
6:         Select boxes $B_i$ where $\text{Score}(B_i) \geq \max(\text{Scores}) - 0.20$
7:     **else**
8:         Select all boxes in the group
9:     **end if**
10:     **if** predicted prompts are disjoint **then**
11:         Classify as MISS
12:     **else**
13:         **if** IoU with any ground truth $\geq 0.85$ **then**
14:             Classify as MATCH
15:         **else**
16:             Classify as MISS
17:         **end if**
18:     **end if**
19: **end for**

---

### 3.2.3 Semantic Stability

Subtle semantic changes in prompts can often lead to varying detections. A semantically robust model should exhibit minimal variation in its predictions for synonymous prompts. To measure semantic variations in our evaluation, we implemented a prompt generation pipeline that creates a series of semantically synonymous sentences for each unique label combination in our ground truth.

Let $I$ be the set of images, $G$ be the set of groups of synonymous prompts, and $P_g$ be the set of synonymous prompts in group $g$. For each image $i \in I$ and group $g \in G$, we first calculate a *Semantic Inconsistency* score for image $i$ and group $g$ as:

$$\text{SI}_{i,g} = std\left(\{C_{i,p} : p \in P_g\}\right), \tag{2}$$

where $std$ represents the standard deviation and $C_{i,p}$ is the confidence score for the predicted box for prompt $p$ on image $i$. Note that the higher the variance of the confidence scores across synonymous prompts, the lower the semantic consistency, i.e., the higher the $\text{SI}_{i,g}$ values.

Finally, the *Semantic Stability* (S) is defined by the the average semantic inconsistency across all images and groups:

$$S = 1 - \frac{1}{|I| \cdot |G|} \sum_{i \in I} \sum_{g \in G} \text{CC}_{i,g}, \tag{3}$$

where $|I|$ is the total number of images, and $|G|$ is the total number of prompt groups.

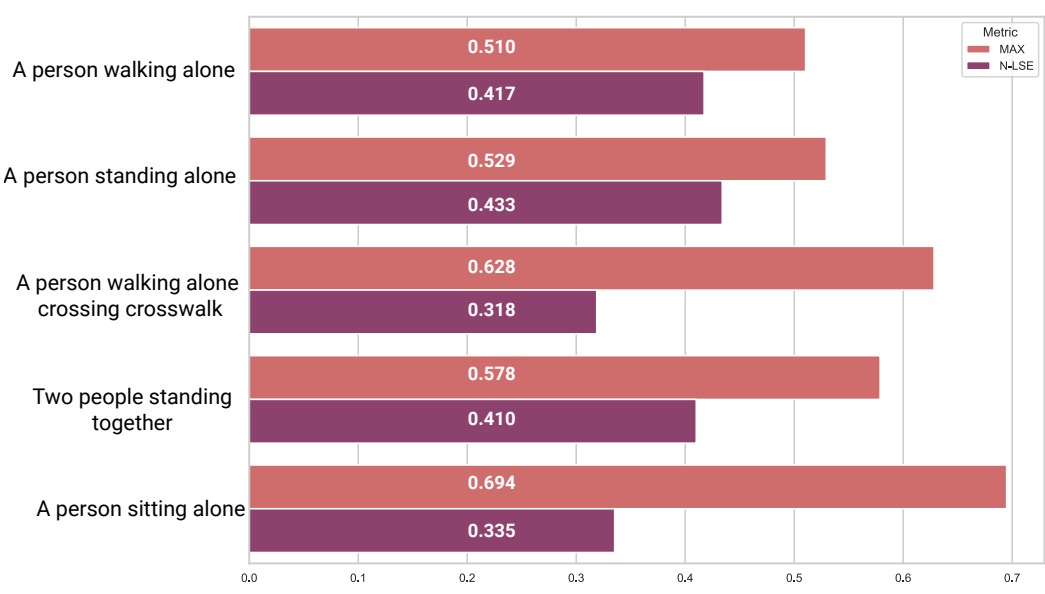

Figure 4: Comparison of average score of the five most frequent prompts computed using the Max-logit and N-LSE (ours). The plot shows how Max-Logit scores can be artificially inflated.

## 4   Results

In this section, we present the findings from our evaluation strategies applied to the ELSA dataset. The ELSA dataset is specifically designed to evaluate the capabilities of open vocabulary detection (OVD) models, and for this purpose, we conducted our experiments using a state-of-the-art OVD model, Grounding DINO. This chapter provides a comparison between our re-ranking with N-LSE and the Max-Logit approach dominantly used in previous work [27, 6, 30]. Furthermore, we highlight the differences in localization performance and provide semantic stability evaluations. In the supplementary material, we present qualitative results showcasing the performance of Grounding DINO on the ELSA dataset.

### 4.1   N-LSE Re-ranking Effects

Grounding DINO has a limit of 900 predictions per image. For our dataset, comprising 924 images, we retrieved all 900 bounding boxes per image and applied a total of 917 prompts to each image. This process generated a substantial total of 762,577,200 bounding boxes. After computing the N-LSE score for all boxes, we retained only those with scores higher than 0.3 (following [32]), resulting in 387,544 predicted boxes, which is equivalent to the 0.05% of the original set of predicted boxes. In contrast, using the Max-Logit method with the same threshold of 0.3 yielded 2,489,685 boxes, approximately 0.3% of the total boxes, which is nearly six times more. This comparison underscores the effectiveness of the N-LSE scoring approach in significantly reducing the number of retained bounding boxes while maintaining high confidence.

Moreover, we computed the average score for each prompt group (i.e., all synonymous prompts) and compared it with the average Max-Logit method. Results show that Max-Logit is often inflated and does not represent the true confidence of the model in multi-label scenarios. We report the comparison for five most frequent prompts in Figure 4. We can observe that the values obtained by the Max-Logit approach are often arbitrarily high, which subsequently, leads to a larger number of false positives.

## 4.2 Localization

Table 1 reports the localization evaluations using the mean average. We choose an higher ranges of threshold due to the high proximity of our bounding boxes: [0.75 - 0.9] with a 0.5 interval. when re-ranking with our N-LSE approach in comparison to the Max-Logit approach. As can be seen, Max-Logit, in general, yields smaller mAP values, since it does not consider all tokens in the prompt, but rather grounds the localization only on the single token with the maximum logit value. In contrast, when using N-LSE, logits of all tokens contribute to the score, and therefore, the model can ground the localization based on all tokens. Consideration of all tokens is crucial when querying for multi-labeled objects in images, for which the model needs to detect the objects based on multiple associated labels.

| Method | CS | CSA | CSAO |
|---|---|---|---|
| Grounding DINO (N-LSE) | 30.4% | 32.4% | 31.1% |
| Grounding DINO (Max-Logit) | 28.9% | 29.7% | 29.3% |

Table 1: mAP (IoU) on ELSA dataset for the different sub-categories when computing the confidence with our ranking method, and the maximum token. *CS* stands for Condition and State, *CSA* also includes Activities, *CSAO* has includes all the categories as descibed in 3

## 4.3 Semantic Stability

Table 2 summarizes the results of our semantic stability measurements when using N-LSE re-ranking in comparison to the Max-Logit approach. Our results show that using N-LSE approach results in up to a 7, 9 and 8 point improvement of the semantic stability when using Grounding DINO on *CS*, *CSA* and *All* categories respectively. Since N-LSE considers the logits of all tokens in the query, it is able to capture the semantics of the entire sequence-level query much better than the Max-Logit approach, and therefore, is more semantically stable across synonymous prompts.

| Method | CS | CSA | All |
|---|---|---|---|
| Grounding DINO (N-LSE) | 0.64% | 0.65% | 64% |
| Grounding DINO (Max-Logit) | 0.57% | 0.56% | 56% |

Table 2: Semantic Stability metric, computed for confidence scores using the default and our N-LSE scoring. *CS* stands for Condition and State, *CSA* also includes Activities, *CSAO* has includes all the categories as descibed in 3

## 5 Conclusion

This paper introduces ELSA, a novel dataset specifically curated for the detection of social activities from still images within urban environments. Employing a multi-labeling scheme, ELSA comprises 924 annotated images, and more than 4,300 bounding boxes, annotated with 115 unique combinations of social activities. ELSA comes with a new re-ranking approach, specifically designed for multi-label scenarios and open vocabulary detection (OVD) models, for which the effect of each token in a query is accounted for in the final confidence score, rather than just the maximum value as in prior work. We demonstrate the success of this approach by adapting a state-of-the-art OVD model to operate on ELSA, showing better performance and more semantic stability across different synonyms.

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
