# OpenReview forum: "ELSA: Evaluating Localization of Social Activities in Urban Streets using Open-Vocabulary Detection"
_NeurIPS.cc/2024/Datasets_and_Benchmarks_Track — Submitted to NeurIPS 2024 Track Datasets and Benchmarks_

### Official Review · Reviewer_Tsa4 · 2024-07-22
**A new benchmark dataset and a novel re-ranking score**

**Rating:** 7
**Confidence:** 3

**Review:**

The main contributions of the paper are twofold: 1) the introduction of a new benchmark dataset, and 2) the development of a novel re-ranking score, N-LSE. The authors dedicate significant attention to explaining N-LSE and demonstrating its advantages over Max-Logit.

The paper only evaluates Grounding DINO on the proposed dataset and discusses the superiority of N-LSE using Grounding DINO. It would be beneficial to include evaluations of other state-of-the-art methods on the dataset and compare the performance of N-LSE with Max-Logit across other multi-object OVD datasets.

Section 4.1: Is it reasonable to choose a threshold of 0.3, as done in [32], and compare the number of remaining bounding boxes? While the metric comparison results are shown in Fig. 4, it might be helpful to include a brief plot or a simple statement indicating that maybe approximately 95% of the remaining bounding boxes are retained after Max-Logit filtering, or something similar.

Is it legal to use Bing and Google street view images (without including actual PNG image files in the dataset, but only providing acquisition codes and annotations) to build a publicly available dataset? Even if it's legal, are there any ethical concerns regarding the use of these images? For example, it appears that the face recognition systems of Google and Bing may have blurred character faces incorrectly, as seen in Fig. 3-a. Have the authors checked the dataset to ensure no faces were missed by the blurring detection, and have they considered the ethical implications of this?

**Strengths:**

The paper proposes a dataset for OVD models to recognize and localize human activity in urban streets from still images, addressing the lack of multi-label benchmarks for social interactions and facilitating related research.

This paper designs a new re-ranking score, N-LSE, which is proven outperforming Max-Logit on the proposed dataset.

**Additional Feedback:**

n/a

**Clarity:**

This paper is well written.

Minor typos:

Line 129, Line 135, Line 139: Use Appendix 6.0x uniformly instead of Section

Line 139: The list of sanity rules are provided in 6.4, not in 6.3

Figure 3: Level "state", not "satate"

Table 2: "All" category has includes all ..., not CSAO

**Correctness:**

Although the dataset downloader code doesn't require API keys, I still can't download the online Street View images, which is probably due to network limitations.

The GitHub repository provides the dataset download code, environment setup commands, and evaluation code.

**Documentation:**

This paper is a sufficient description of the proposed dataset. README file are included in the dataset link. Since the dataset is organized in the baseline format, the baseline reproduction code should be the original baseline code.

**Ethics:**

The author bears all responsability for any potential violations of rights.

**Limitations:**

The author discusses the limitations of the work in Appendix A.7.

**Opportunities For Improvement:**

Including more comparison datasets and baseline methods, as suggested in the review section above, would be beneficial.

**Relation To Prior Work:**

Yes. The author introduces the previous efforts on social interactions in public spaces, open-vocabulary object detection (OVD), OVD evaluation, and activity localization datasets.

**Summary And Contributions:**

Due to the lack of multi-label benchmarks for social interactions, this paper proposed ELSA to evaluate the performance of OVD models in recognizing and localizing human activity in urban streets from still images. ELSA contains more than 4,000 bounding boxes annotated with 115 unique combination of human activities for 900 street view images. Furthermore, N-LSE, a novel re-ranking score, was designed, and overcomed Max-Logit on effectiveness. Besides, this paper proposed a confidence-based dynamic box aggregation algorithm, inorder to handle multiple detected predictions of the same object.

---

> ### Author Rebuttal · Authors · 2024-08-17
>
> Thank you for the insightful comments and close attention to details. We very much appreciate the raised points.
>
> > It would be beneficial to include evaluations of other state-of-the-art methods on the dataset and compare the performance of N-LSE with Max-Logit across other multi-object OVD datasets.
>
>  Thank you for the comment. As mentioned in the main section – we are currently generating predictions with new models and one is already integrated into the pipeline. We plan to report those results during the discussion period.
>
> > While the metric comparison results are shown in Fig. 4, it might be helpful to include a brief plot or a simple statement indicating that maybe approximately 95% of the remaining bounding boxes are retained after Max-Logit filtering, or something similar.
>
> we also provided this information in the original submission, in Section 4.1. To make it easier for you, here is the excerpt where such information is detailed. If our understanding of your concern is not correct, we appreciate you clarifying the points so we can best address it.
>
> *After computing the N-LSE score for all boxes, we retained only those with scores higher than 0.3 (following[32]), resulting in 387,544 predicted boxes, which is equivalent to the 0.05\% of the original set of predicted boxes. In contrast, using the Max-Logit method with the same threshold of 0.3 yielded 2,489,685 boxes, approximately 0.3\% of the total boxes, which is nearly six times more. This comparison underscores the effectiveness of the N-LSE scoring approach in significantly reducing the number of retained bounding boxes while maintaining high confidence.*
>
> > Is it legal to use Bing and Google street view images (without including actual PNG image files in the dataset, but only providing acquisition codes and annotations) to build a publicly available dataset?
>
> According to previous work in the field, this approach aligns with legal standards as long as the images themselves are not distributed. Similar practices have been employed in other publicly available datasets such as [CVUSA](https://mvrl.cse.wustl.edu/datasets/cvusa/), [City Street View Dataset](https://www.kaggle.com/datasets/stelath/city-street-view-dataset), and [GMCP Geolocalization](https://www.crcv.ucf.edu/data/GMCP_Geolocalization/), among others, including Street Score, which we have cited in our Datasheet.
>
> > Even if it's legal, are there any ethical concerns regarding the use of these images? For example, it appears that the face recognition systems of Google and Bing may have blurred character faces incorrectly, as seen in Fig. 3-a. Have the authors checked the dataset to ensure no faces were missed by the blurring detection, and have they considered the ethical implications of this?
>
> We very much appreciate your concern and share the same commitment to ensuring privacy and ethical standards in our research. The privacy of individuals captured in street view images is of utmost importance to us. The blurring of faces, including those on advertisements, is a testament to the effectiveness of the anonymization protocols implemented by Bing and Google. While the primary responsibility for face blurring lies with these providers, we understands the possibility of False negatives and failure of the face recognition models and our team takes this issue very seriously. We have manually annotated the dataset with a focus on pedestrian subjects, and we have taken great care to verify that all faces in the images are properly anonymized. This meticulous review reflects our dedication to upholding the highest privacy standards in our work.
>
> > Although the dataset downloader code doesn't require API keys, I still can't download the online Street View images, which is probably due to network limitations.
>
> Regarding the speed to download the ELSA images, we are currently working on a faster implementation which makes use of the Google API keys. However, for the scope of the review, we can provide the data to you and the reviewer team with a private link to the imagery.
>
>
> Regarding the clarity points you raised, thank you for the close read. We have resolved these issues and made a revision in the text to ensure the following corrections:
> - **Line 129, Line 135, Line 139**: We have uniformly referred to Appendix 6.0x instead of Section, as suggested.
> - **Line 139**: The reference has been corrected to specify that the list of sanity rules is provided in Appendix 6.4, not in Appendix 6.
> - **Figure 3**: The typo has been corrected, changing "satate" to "state."
> - **Table 2**: Thanks for the attention. The absence of CSAO in the chart was due to a low match rate of CSAO annotations. When a CSAO prediction matches, it is only a true positive if the annotation is identical to it. In contrast, a CS prediction is a true positive if it matches the CS from a CSAO. So we have some CS predictions that matched with CSAO, but no CSAO that matched with other CSAO.
> CSAO comprised 16% of the prompts, 1.2% of the annotations, and 0.26% of the annotations in the matches. Among those exactly 100 CSAO matches, 5 were with CSAO predictions. Only CSAO truth can be a true positive for CSAO predictions, and each of these 5 were false positives. Because there were no true positives, TP was 0, so both precision and recall were 0, which returns np.nan for AP, explaining why CSAO is not included. We didn’t include these details due to the space limit, but if the reviewer suggests, we can include this explanation in the appendix.

---

> > ### Comment · Reviewer_Tsa4 · 2024-08-26
> > **Reply**
> >
> > Thank you for your response.
> >
> > I've updated my score. Please let me know when you add new experiments.
> >
> > Regarding Table 2, I want to ensure consistency between the classes listed in the table ("CS, CSA & All") and those described in the caption ("CS stands for ..., CSA also ..., CSAO has ...").

---

> > > ### Author Rebuttal · Authors · 2024-08-30
> > >
> > > Thank you for the positive comments, and constructive feedback and suggestions which lead to much richer analysis, and motivated some interesting future research in our team.
> > >
> > > We have now added additional analysis and experiments to the main rebuttal.

---

### Official Review · Reviewer_jcjB · 2024-07-25

**Rating:** 5
**Confidence:** 3
**Correctness:** It appears that there are no concerns…
**Clarity:** It appears that there are no concerns…

**Review:**

- The task discussed in the paper appears to be original and challenging for the model to recognize, and it seems to have potential contributions to specific industries.
- The clarity and articulation of the proposal are also pronounced.

**Strengths:**

- The design idea of constructing a multi-label dataset that contributes to the scientific basis of urban planning and prediction, referencing the correlation between street characteristics and fixed social behaviors from the perspectives of urban sociology and design, is intriguing.
- A quantitative (mAP) and qualitative (semantic) comparative analysis applies the CDBA for handling predictions detected multiple times on the same object and the N-LSE metric, which considers the logits of all tokens within a query.

**Additional Feedback:**

If there are any plans or visions for maintaining and expanding the dataset in the future, mentioning them would likely help promote the interest and value of this benchmark to readers.

**Documentation:**

The proposed method discusses the composition and organization of the dataset. However, it does not appear to address maintenance or expansion.

**Ethics:**

It appears that there are no concerns regarding ethics.

**Limitations:**

- In lines 43-46, it seems that the need for data is emphasized because the development of robust models that can be generalized to various urban scenarios is hindered. However, considering the amount of data and the configuration of annotations, it remains questionable whether these issues can be effectively addressed.
- As mentioned in the discussion, the evaluation is conducted using only a single model, which leaves the assessment of the semantic stability of the evaluation metrics somewhat ambiguous and inadequately supported by data.
- The application and comparison of evaluation metrics that emphasize semantic stability and reliability based on all tokens within a query are interesting. However, it raises doubts whether the high performance of these proposed metrics can indeed underscore the necessity and excellence of this benchmark dataset.
- After line 95, in the discussed similar activity location datasets, it is disappointing that there is no comparison with existing datasets regarding social activities (like sitting, standing, or other poses and activities). Demonstrating how the proposed dataset includes different issues compared to existing ones is crucial in dataset evaluation.

**Opportunities For Improvement:**

none

**Relation To Prior Work:**

The paper somewhat addresses how it differentiates from prior research in the problem domain it seeks to address.

**Summary And Contributions:**

- Based on the theoretical frameworks of urban sociology and urban design, multi-label detection data on social interaction activities is collected, and the OVD model is applied and evaluated for this task.
- A new evaluation protocol is proposed that adjusts semantic stability and predictive reliability, and it is compared with existing methods.

---

> ### Author Rebuttal · Authors · 2024-08-17
>
> Thank you for the great points raised.
>
> > In lines 43-46, it seems that the need for data is emphasized because the development of robust models that can be generalized to various urban scenarios is hindered. However, considering the amount of data and the configuration of annotations, it remains questionable whether these issues can be effectively addressed.
>
> Thank you for the comment. We could have stated this more clearly. Here, we wanted to emphasize the role of benchmark and advanced evaluation metrics in uncovering the vulnerabilities and limitations of these models, which, in turn, informs their future development. Similarly ELSA exposes critical shortcomings in the ability of the SOTA OVD models in accurately capturing social and individual action. We believe that ELSA represents a pivotal step towards the effective evaluation of OVDs in urban scenarios. Our current version of ELSA dataset is designed exclusively for evaluation and is, as correctly mentioned, not intended for training purposes.
> To address this concern, we have revised this sentence to more clearly reflect the intended meaning.
>
> Revised sentence: *The absence of benchmark data for open-vocabulary detection of social and individual actions in still images 'in the wild' hinders the development of robust models that generalize well across diverse and spontaneous urban scenarios, where the context and variability of human activities are far greater than those typically encountered in controlled environments, and where the role of benchmarks and advanced evaluation metrics is critical in uncovering the vulnerabilities and limitations of these models, informing their future development.*
>
> > As mentioned in the discussion, the evaluation is conducted using only a single model, which leaves the assessment of the semantic stability of the evaluation metrics somewhat ambiguous and inadequately supported by data.
>
> Regarding the experiments on additional models, – as mentioned in the main section – we are currently generating predictions with new models and one is already integrated into the pipeline. We plan to report those results during the discussion period.
>
> > After line 95, in the discussed similar activity location datasets, it is disappointing that there is no comparison with existing datasets regarding social activities (like sitting, standing, or other poses and activities). Demonstrating how the proposed dataset includes different issues compared to existing ones is crucial in dataset evaluation.
>
> With respect to providing a more detailed comparison with other datasets, we appreciate the suggestion and fully agree with its merit. However, given the limited time frame, we had to carefully prioritize our tasks. In this instance, our primary focus was on adding more models to showcase the value of ELSA benchmark dataset and the robustness and merits of our suggested metrics. While we recognize the importance of the proposed addition, it was necessary to prioritize our current objectives within the constraints of the time available. If the reviewers find this addition to be critical, we will commit to working on providing the table before the end of the discussion period.

---

### Official Review · Reviewer_nrdp · 2024-07-25
**An interesting benchmark for evaluating the localization of social activities in urban streets by open-vocabulary detector**

**Rating:** 6
**Confidence:** 4
**Correctness:** Yes
**Clarity:** Yes

**Review:**

The paper is well-written and easy to understand. Localizing social activities has many real-world applications, which makes the present paper well-motivated. The author also proposes a novel dataset, re-ranking method, and evaluation techniques to study this new research problem. The proposed re-ranking technique and the confidence-based dynamic box aggregation are relevant and sound.

**Strengths:**

Please refer to the Review section for more details.

**Additional Feedback:**

N/A.

**Documentation:**

Yes

**Ethics:**

N/A.

**Limitations:**

Please refer to the Opportunities For Improvement section for more details.

**Opportunities For Improvement:**

1. The number of benchmarked detectors is insufficient. The author only benchmarked the Grounding DINO in the present paper, while there are lots of OVD that can be considered. The author may want to add more OVD to construct a more comprehensive benchmark.

2. The hyperparameters in CDBA are pre-defined, and it is hard to know whether they may influence the evaluation results. The author may want to provide more ablation studies to show the impact of the selected thresholds in CDBA.

**Relation To Prior Work:**

Yes

**Summary And Contributions:**

This paper studies the problem of localizing social activities in urban streets using the open-vocabulary detector (OVD). The author proposes a novel multi-label dataset tailored to this problem for evaluation, a novel re-ranking method for predictive confidence, and new evaluation techniques for OVD.

---

> ### Author Rebuttal · Authors · 2024-08-17
>
> Thank you for close attention and constructive comments.
> > The author may want to add more OVD to construct a more comprehensive benchmark.
>
> Regarding the experiments on additional models, – as mentioned in the main section – we are currently generating predictions with new models and one is already integrated into the pipeline. We plan to report those results during the discussion period.
>
> > The hyperparameters in CDBA are pre-defined, and it is hard to know whether they may influence the evaluation results. The author may want to provide more ablation studies to show the impact of the selected thresholds in CDBA.
>
> We also agree with the need for CDBA to be parameterized. In the attached PDF to the main rebuttal, we have included a new version of the algorithm which includes parametrization. We are extending our codebase to accept the threshold as a hyper-parameter. In future works using our new metrics and algorithms, we plan to do a full ablation study to examine the impact of the confidence score range on the different evaluation metrics such as mAP, and different variations of F1.

---

### Author Rebuttal · Authors · 2024-08-17

We wish to thank all reviewers for their invaluable efforts in reviewing our manuscript.

As a summary of the reviews, we are pleased to see that the reviewers found our paper to be well-written [`nrdp`, `Tsa4`], clear, and easy to understand [`nrdp`, `jcjB`]. The reviewers also recognized the value and contribution of the dataset [`nrdp`, `Tsa4`, `jcjB`], as well as the novelty of the proposed N-LSE re-ranking method [`nrdp`, `Tsa4`,`jcjB`]. Additionally, the semantic stability metric [`jcjB`] and CDBA algorithm [`nrdp`, `jcjB`, `Tsa4`] were appreciated by the reviewer. Concerns raised include the evaluation being done only on one model [`nrdp`, `Tsa4`, `jcjB`], alignment of the motivation of the paper and the size of the dataset [`jcjB`], parameterization of CDBA [nrdp], comparison of the dataset attributes with existing human activity detection datasets [`jcjB`, `Tsa4`], privacy concerns [`Tsa4`], and plans for extension and maintenance[`jcjB`].

We address these concerns in the revision and individual responses below. Major changes are as follows:

1. **CDBA Enhancements**: In response to the concerns regarding CDBA, we have revised and improved our algorithm definition and declaration, and integration. The revised algorithms are presented in the attached PDF file. In summary:
- We have parameterized CDBA, making it more flexible to be used in different scenarios and with various models.
- To increase the clarity of how CDBA is integrated into the evaluation process, we have provided a new algorithm, showing in detail how this integration is being implemented.
- We plan to add a new table to our result section, showing the results with and without CDBA integration.

2. **Optimized Codebase and Workflow Enhancements**: The codebase has been extensively re-engineered to streamline the user experience, automating weight downloads, project setup, and input file management, while eliminating the need for manual path specifications—except for image files. We've also tackled the complexities of model-specific installations and dependency conflicts by embedding all configurations within a unified installation process, enabling seamless model integration.
Comprehensive notebooks and visualizations have been added, detailing each step of the workflow. Additionally, we've enhanced the ability to refine or subset which images or prompts are processed during inference. Improved documentation and inline comments further enhance code readability and usability. We hope these enhancements improves the reproducibility and show our commitment to support and maintain ELSA.

3. **Addition of new OVD models**: Reviewers have suggested a comparison with additional OVD models. We are currently generating prediction scores with two more models and have already implemented one in our pipeline. We plan to post the results of these experiments within the next few days during the discussion period.
We would also like to highlight the scale of our evaluation. Using 917 prompts across 924 images, we retained 900 bounding boxes per image, resulting in a benchmark involving over *762 million boxes*. These boxes were re-ranked using our proposed NLSE score, then thresholded and processed through CDBA, after which the final evaluation metrics were computed. While we fully acknowledge this limitation, it is important to note that our approach pushes beyond the boundaries of conventional benchmarking. The scale of our evaluation challenges the existing norms, offering insights that extend the capabilities of traditional methods.

---

> ### Author Rebuttal · Authors · 2024-08-30
>
> Thank you for your valuable feedback and recognition. In response, we have expanded our analysis to **include two additional variations of the MDETR model**—one with an EfficientNet backbone and another with a ResNet-101 backbone—as well as **three variations of Grounding DINO** – with SwinT and SwinB backbones – pre-trained on different datasets.
>
> **Our CDBA and Semantic Stability metrics are inherently model-agnostic**, ensuring broad applicability across various architectures. Additionally, **our NLSE box re-ranking is specifically tailored for models with phrase grounding capabilities**, where there is a direct alignment between query tokens and box confidence scores, enhancing interpretability and precision in detection outcomes.
>
> During our review, we identified and corrected a bug in the mAP calculations and adjusted the mAP of models, now matching our qualitative results, as we expected the accuracy to be much lower than the mAP values in our initial results. After addressing this issue, we conducted extensive unit tests to ensure the accuracy and reliability of the corrected outputs. Importantly, **this does not affect our core argument, which remains valid**; the correction only adjusted the scale of mAP scores, not the observed trends and conclusions.
> The updated findings, presented in two tables, provide a more precise evaluation of the models. We report the mAP for each interaction level (CS, CSA, CSO).
>
>
> Table 1 illustrates the impact of **our re-ranking method, NLSE, in comparison with Maximum Confidence** of the  **conventional mAP** . We chose this mAP metric to isolate the effect of each confidence computation metric. As expected, NLSE effectively reduces false positives from over-estimated confidence scores, commonly associated with max confidence calculations, leading to improved mAP results across all baselines and levels. The mAP scores are computed at four levels: global, Condition + State (CS), Condition+State+Activity (CSA) and Condition+State+Other (CSO)  Scores are in percentage.
>
> | Model | Variation    | Ranking | global       | cs           | csa          | cso          |
> |-------|--------------|---------|--------------|--------------|--------------|--------------|
> | Gdino | Swin-B       | NLSE    | **2.32E-01** | **6.76E-01** | **1.89E-02** | **3.20E-03** |
> |       |              | Max     |     1.99E-01 |     5.16E-01 |     9.90E-03 |     1.80E-03 |
> |       | Swin-T (1)   | NLSE    | **9.69E-02** | **1.90E-01** | **3.25E-03** | **8.00E-04** |
> |       |              | Max     |     9.63E-02 |     1.81E-01 |     3.20E-03 |     6.00E-04 |
> |       | Swin-T (2)   | NLSE    | **1.00E-01** | **2.56E-01** | 1.10E-03 | **4.30E-03** |
> |       |              | Max     |     9.58E-02 |     2.24E-01 |     **3.60E-03** |     5.00E-04 |
> | MDETR | EfficientNet | NLSE    | **1.63E-04** | **8.00E-04** | **6.60E-05** | **6.00E-06** |
> |       |              | Max     |     1.00E-06 |     3.00E-06 |     0 |     0 |
> |       | ResNet 101   | NLSE    | **7.80E-05** | **4.02E-04** | **2.50E-05** | **4.00E-06** |
> |       |              | Max     |            0 |            0 |            0 |            0 |
>
>
> Table 2 compares the **performance of NMS-AP and CDBA-AP** using NLSE confidence metrics.
>
>
> | Model | Variation    | Scoring | global   | cs       | csa      | cso      |
> |-------|--------------|---------|----------|----------|----------|----------|
> | Gdino | Swin-B       | CDBA-AP | 1.40E-01 | 3.81E-01 | 6.27E-03 | 2.42E-03 |
> |       |              | NMS-AP  | 4.44E-01 | 1.04E+00 |        0 | 2.51E-02 |
> |       | Swin-T (1)   | CDBA-AP | 9.32E-03 | 1.87E-02 | 5.74E-04 | 4.23E-04 |
> |       |              | NMS-AP  | 5.08E-01 | 2.27E+00 | 3.18E-02 |        0 |
> |       | Swin-T (2)   | CDBA-AP | 3.41E-02 | 8.94E-02 | 9.84E-05 | 1.38E-03 |
> |       |              | NMS-AP  | 2.10E-01 | 3.47E-01 |        0 |        0 |
> | MDETR | EfficientNet | CDBA-AP | 4.20E-05 | 2.09E-04 | 1.60E-05 | 1.59E-06 |
> |       |              | NMS-AP  | 0        | 0        | 0        | 0        |
> |       | ResNet 101   | CDBA-AP | 1.05E-05 | 5.45E-05 | 3.38E-06 | 4.40E-07 |
> |       |              | NMS-AP  | 0        | 0        | 0        | 0        |
>
>
> Overall, **Grounding DINO significantly outperforms MDETR** on our task. MDETR's method of computing box confidences results in consistently higher confidence scores per box compared to Grounding DINO, yet yields a substantially lower count of true positive detections. **Grounding DINO's final evaluation aligns with expectations**; consistent with the observations from the NMS-AP paper, our **CDBA approach mitigates the issue of inflated AP scores**, providing a more accurate assessment of model performance by producing lower but more representative AP values.
> However, a notable trend emerges with MDETR: nearly all classes show an AP of zero when evaluated with NMS-AP, highlighting the vulnerability of NMS-AP that was also identified in the paper—where, in suboptimal models and challenging cases, the highest confidence score does not correspond to the correct prediction. The results from both standard and CDBA-AP evaluations reinforce this shortcoming, demonstrating that **our CDBA-AP evaluation method can recover a number of true positive predictions**, leading to non-zero AP scores and a more reliable measure of the model’s performance.
>
> This comprehensive evaluation offers a clearer understanding of how these models perform under different conditions, highlighting the robustness and limitations of current approaches in detecting complex social interactions while affirming the validity of our conclusions.

---

> > ### Author Rebuttal · Authors · 2024-08-30
> >
> > To provide a complete picture, Table 3 reports the same evaluation of Table 2 using the **maximum confidence score** instead of our ranking algorithm. The results are consistent with what we described above.
> >
> >
> > | Model | Variation    | Scoring | global   | cs       | csa      | cso      |
> > |-------|--------------|---------|----------|----------|----------|----------|
> > | Gdino | Swin-B       | CDBA-AP | 3.48E-02 | 9.76E-02 | 2.03E-03 | 1.17E-03 |
> > |       |              | NMS-AP  | 7.50E-01 | 2.31E+00 | 5.45E-02 | 5.24E-01 |
> > |       | Swin-T (1)   | CDBA-AP | 6.87E-03 | 1.33E-02 | 4.83E-04 | 1.60E-04 |
> > |       |              | NMS-AP  | 4.44E-01 | 1.87E+00 | 3.63E-03 |        0 |
> > |       | Swin-T (2)   | CDBA-AP | 8.76E-03 | 2.32E-02 | 7.51E-04 | 7.96E-05 |
> > |       |              | NMS-AP  | 2.19E-01 | 4.17E-01 | 1.06E-02 | 5.32E-02 |
> > | MDETR | EfficientNet | CDBA-AP | 1.54E-07 | 7.52E-07 | 3.00E-09 | 4.00E-09 |
> > |       |              | NMS-AP  |        0 |        0 |        0 |        0 |
> > |       | ResNet 101   | CDBA-AP | 1.30E-08 | 3.20E-08 |        0 |        0 |
> > |       |              | NMS-AP  |        0 |        0 |        0 |        0 |

---

### Decision · Program_Chairs · 2024-09-26

**Decision:**

Reject

**Comment:**

I would like to congratulate the authors for their great work and the considerable effort put into the rebuttal.

The reviewers seemed pretty satisfied from the very beginning with this paper. Also, the final review ratings have been mostly positive, receiving scores of 6, 5, and 7. In fact, the concerns of the reviewer who rated the paper a 5 were addressed during the review process, and although we did not receive a response from them, I believe they would be quite satisfied.

The authors themselves have already highlighted the paper's positive aspects and delivered on their promises (e.g., additional experiments with backbones, parameterization, etc.). This is high-quality work that deserves recognition. The paper is extremely well-written, the codebase appears solid, and it is original in its proposal of a new re-ranking algorithm, evaluation metric, and significant new insights into the OVD problem that will help advance the field further.

Overall, I recommend this paper for acceptance.

Note from PC: This year, the track has been incredibly competitive, which meant that many good papers had to be rejected. After careful discussion with the SACs we have concluded that this paper unfortunately cannot be accepted this time. This is the final decision, which cannot be appealed. We encourage the authors to incorporate feedback from reviewers and additional results / discussion provided during the author response period in their next submission.